# Extended-spectrum β-lactamase-producing *Escherichia coli* isolated from captive primates: characteristics and horizontal gene transfer ability analysis

**Wenhao Zhong[1], Yuxin Zhou[1], Mengjie Che[1], Liqin Wang[2], Xingyu Tian[1], Chengdong Wang[3], Yuehong Cheng[4], Haifeng Liu[1], Ziyao Zhou[1], Guangneng Peng[1], Kun Zhang[1], Yan Luo[1], Keyun Shi[5]\*, Zhijun Zhong [1]\***

**1** College of Veterinary Medicine, Sichuan Agricultural University, Key Laboratory of Animal Disease and Human Health of Sichuan, Chengdu, China, **2** The Chengdu Zoo, Institute of Wild Animals, Chengdu, China, **3** China Conservation and Research Centre for the Giant Panda, Key Laboratory of SFGA on The Giant-Panda, Chengdu, Sichuan, China, **4** Sichuan Wolong National Natural Reserve Administration Bureau, Wenchuan, China, **5** Jiangsu Yixing People's Hospital, Yixing, China

☯ These authors contributed equally to this work
\* zhongzhijun488@126.com (ZZ); staff955@yxph.com (KS)

## Abstract

The rapid spread of extended-spectrum β-lactamases (ESBLs)-producing *Escherichia coli* (ESBL-EC) around the world has become a significant challenge for humans and animals. In this study, we aimed to examine the characteristics and horizontal gene transfer (HGT) capacity of ESBL-EC derived from captive primates. We screened for ESBL-EC among a total of 444 multidrug-resistant (MDR) *E. coli* strains isolated from 13 zoos in China using double-disk test. ESBL genes, mobile genetic elements (MGEs), and virulence-associated genes (VAGs) in ESBL-EC were detected through polymerase chain reaction (PCR). Furthermore, conjugation experiments were conducted to examine the HGT capacity of ESBL-EC, and the population structure (phylogenetic groups and MLST) was determined. Our results showed that a total of 69 (15.54%, 69/444) ESBL-EC strains were identified, and 5 variants of $bla_{CTX}$ and 3 variants of $bla_{TEM}$ were detected. The highest detection rate was $bla_{CTX-M-55}$ (49.28%, 34/69), followed by $bla_{CTX-M-15}$ (39.13%, 27/69). Ten MGEs were detected and the most prevalent was *IS26* (78.26%, 54/69), followed by *ISEcp1* (60.87%, 42/69). Eighteen combinations of MGEs were detected, in which *ISEcp1+IS26* was predominant (18.84%, $n = 13$). A total of 15 VAGs were detected and the most prevalent was *fimC* (84.06%, 58/69), followed by *sitA* (78.26%, 54/69). Furthermore, HGT ability analysis results showed that 40.58% (28/69) of ESBL-EC strains exhibited the ability to engage in conjugative transfer. Plasmid typing revealed that IncFIB (78.57%, 22/28) had the highest detection rates. Furthermore, antibiotic resistance genes (ARGs) of $bla_{TEM-135}$, *tetA and qnrS*; MGEs of *IS26*, *trbC* and *ISCR3/14* showed high rates of conjugative transfer. The population structure analysis showed that the phylogroup B1 and ST2161 were the most prevalent. ESBL-EC poses a potential threat to captive primates and may spread to other

**Data availability statement:** All relevant data are within the paper and its Supporting Information files.

**Funding:** This work was funded by Study on Key Technologies for Conservation of Wild Giant Panda Populations and Its Habitats within Giant Panda National Park System (CGF2024001), and Chengdu Giant Panda Breeding Research Foundation (CPF2017-05, CPF2015-4).

**Competing interests:** The authors have declared that no competing interests exist.

animals, humans, and the environment. It is imperative to implement measures to prevent the transmission of ESBL-EC among captive primates.

## 1. Introduction

*Escherichia coli* is a resident microorganism in the gut of humans and animals, as well as in the environment [1,2]. Antimicrobial resistant (AMR) *E. coli* poses a threat to animal and human health [3,4]. The production of extended`spectrum β-lactamases (ESBLs) by *E. coli* represents the primary mechanism responsible for the treatment failure by β-lactam antibiotics [5]. The ESBL-producing *E. coli* (ESBL-EC) is one of the most important pathogens at the One Health interface [6]. Current ESBLs are mainly encoded by $bla_{CTX-M}$, $bla_{TEM}$, $bla_{SHV}$ and $bla_{OXA}$ genes. CTX-Ms are the most predominant types of ESBLs which are encoded by several $bla_{CTX-M}$ type genes [7]. According to amino acid homology, $bla_{CTX-M}$ was divided into five subfamilies (CTX-M-1, CTX-M-2, CTX-M8, CTX-M-9 and CTX-M-25 group). More than 260 CTX-M-β-lactamase variants have now been identified [8]. ESBL genes frequently co-exist with other antibiotic resistance genes (ARGs), such as *qnrS*, *tetA* and *sul*, which cause multidrug-resistance (MDR) in *E. coli* [9–11]. ESBL-EC has been frequently reported in captive and wild animals, such as giant pandas, ostriches, white storks, herring gull and Fennec fox [12–16]. However, there have been only few reports focus on the ESBL-EC from captive primates recently. In 2003, 65 ESBL-EC strains (32%) were detected from primates in six zoos in China, among which $bla_{CTX-M-15}$ (48%) was the most predominant [17]. In 2012, one ESBL-EC strain was isolated from a wild Barbary macaque (*Macaca sylvanus*) in Algeria [18]. In 2022, Zeng *et al* [19] isolated eight ESBL-EC strains from captive primates, all of which carried the $bla_{CTX-M}$ genes. In 2024, Bazalar-Gonzales et al [20] isolated seven ESBL-EC strains from captive and semi-captive primates, three $bla_{CTM-X}$ variants ($bla_{CTX-M-15}$, $bla_{CTX-M-55}$ and $bla_{CTX-M-65}$) were identified. Previous studies have focused on the identification of ESBL-EC and the prevalence of ESBL genes. However, other characteristics, such as the carriage of mobile genetic elements (MGEs), virulence-associated genes (VAGs), horizontal gene transfer (HGT) ability and population structures of ESBL-EC isolated from captive primates remain unknown.

 *E. coli* is a reservoir of ARGs and VAGs [21]. The transmission of ESBL genes and ARGs is related to MGEs, including plasmids, transposons and insertion sequences (ISs) [22,23]. ESBL genes are often co-located in plasmids with other ARGs, which cause MDR in the same or different species of bacteria by the horizontal transmission through conjugation transfer [24,25]. Furthermore, transposons can jump from one part of a gene sequence to another with co-transferring of ARGs [26]. Insertion sequences (ISs) have been frequently reported to be associated with the transfer of ESBL genes, for example, *ISEcp1* and *IS26* have been widely reported to be associated with $bla_{CTX-M}$ gene transmission. Therefore, $bla_{CTX-M}$ has rapidly become the most common ESBL genes transferred by MGEs [27,28]. Additionally, plasmids were also reported to carry various VAGs and to be associated with the transfer of VAGs, for example, IncF plasmid was the most reported virulence plasmid [29,30]. Previous studies showed that ESBL-EC also harbored a range of VAGs, which may help to the enhanced fitness of these emerging CTX-M-harboring strains [31,32]. The strains that carry various VAGs may not be harmful to animals, but there is a possibility for them to become pathogens if host immunity weakens or in response to other adverse environmental factors [21]. The ESBL-EC strains which carried VAGs also cause treatment difficulties [33]. Both the ESBL genes and the VAGs can be transferred horizontally through plasmids together, which will pose a more serious threat to the survival of captive animals [33].

In our previous study, we analyzed the AMR of *E. coli* isolated from primates in 13 zoos, and the results demonstrated that the strains exhibited high resistance to β-lactam antibiotics [34]. In this study, we proceeded to identify ESBL-EC in MDR isolates from captive primates and conducted a more detailed analysis of the HGT ability of ESBL-EC for ESBL genes, ARG, VAGs and MGEs.

## 2. Materials and methods

### 2.1 Bacterial strains

Four hundred and forty-four MDR *E. coli* strains from captive primates, which were isolated from 13 zoos and had previously been stored in our laboratory, were used in this study [34].

### 2.2 Screening of ESBL-EC isolates

The third generation cephalosporins (cefotaxime, ceftazidime) alone and combined with clavulanic acid were used to confirm the ESBL phenotype by combined disk method according to the Clinical and Laboratory Standards Institute (CLSI, 2023) guidelines. ESBL-EC strains were screened using cefotaxime (CTX, 30 μg), cefotaxime + clavulanic acid (CTL, 30/10 μg), ceftazidime (CAZ, 30 μg) and ceftazidime + clavulanic acid (CAL, 30/10 μg) antibiotic disk (Oxoid, Basingstoke, United Kingdom). An antibacterial zone ≥ 5 mm in diameter with the combination of two antimicrobials compared with clavulanate alone is considered to be ESBL-producing.

### 2.3 Screening of ESBL genes, VAGs and MGEs from ESBL-EC isolates

Total genomic DNA was extracted using the TIANamp Bacteria DNA kit (Tiangen Biotech, Beijing, China). The DNA samples were stored at − 20 °C. We screened for 8 ESBL genes: $bla_{OXA}$, $bla_{TEM}$, $bla_{SHV}$ and $bla_{CTX-M}$ (group1, 2, 8, 9 and 25). Additionally, 20 VAGs in 5 categories were analyzed, including antiserum survival factor (*cvaC, ompT, iss*), adhesion-related genes (*tsh, eaeA, papA, fimC*), hemolysin-related genes (*hlyA, hlyF*), invasion and toxin-related genes (*elt, estA, estB, vat, ipaH, astA*), and iron transport-related genes (*fyuA, iroN, Irp2, iucD, sitA*). The occurrence of the following MGEs were investigated: *ISEcp1, IS26, IS903, ISCR1, ISCR3/14, traA, trbC, traF, tnpU, tnpA/Tn21, tnsA, merA, tnp513, IS1133, ISpa7, ISaba1, ISkpn6* and *ISkpn7*. The sequences and conditions for the PCR primers were presented in S1 Table. PCR products were separated by gel electrophoresis in a 1.0% agarose gel stained with GoldView™ (Sangon Biotech, Shanghai, China), visualized under ultraviolet light and photographed using a gel documentation system (BioRad, Hercules, USA). The positive products were sequenced using Sanger sequencing by Tsingke Biotech (Beijing, China), and the sequences were analyzed online using the BLAST function of NCBI (http://blast.ncbi.nlm.nih.gov).

### 2.4 Conjugation experiment and plasmid incompatibility groups

In this study, the sodium azide-resistant *E. coli* J53 was used as the recipient strain to evaluate the conjugation and transfer ability of ESBL-EC. Those colonies capable of growing on Luria-Bertani (LB) agar supplemented with CTX (4 mg/L) and sodium aside (100 mg/L) were considered transconjugants. All transconjugants were tested for ESBLs phenotype, and the presence of ESBL genes in transconjugants was verified by PCR and DNA sequencing. Select the culture plates at appropriate dilution gradients for colony counting (where the number of colonies ranges from 30 to 300). Transfer frequency was determined by calculating the ratio of the number of transconjugants to the number of recipient cells involved in the experiment

[35]. In our previously published research, we had already detected the ARGs carried by the donor bacteria, including those for tetracyclines (*tetA*, *tetG*), sulfonamides *(sul1*, *sul2*, *sul3*), quinolones/fluoroquinolones (*qnrS*, *oqxAB*), chloramphenicol/phenicol (*flor*, *cmlA*, *cat1*), and aminoglycosides (*aacC2*, *aphA3*) [34]. We then examined whether the transconjugants carry these same genes. In addition, VAGs, and MGEs were detected in transconjugants. Plasmid incompatibility groups were detected in bacteria (donor) and their transconjugants as described previously [36]. In brief, eighteen pairs of primers were designed to recognize K, B/O, FrepB, FIB, W, I1, FIA, FIIA, N, T, HI2, X, Y, FIC, A/C, L/M, P and HI1. Primers and reaction conditions were listed in S1 Table.

## 2.5 Phylogenetic grouping and MLST

ESBL-EC strains with conjugative transfer ability were selected for this experiment. Phylogenetic analysis (A, B1, B2, D) was conducted by detecting *yjaA*, *chuA*, and *TSPE4.C2,* according to Clermont et al [37]. MLST was performed according to pubMLST recommended methods (https://pubmlst.org/organisms/escherichia-spp). Briefly, seven housekeeping genes (*purA*, *recA gyrB*, *icd*, *mdh*, *adk* and *fumC*) were investigated by PCR, the obtained DNA fragments were sequenced and uploaded to the website (https://pubmlst.org) to obtain the allele number, and the sequence type (ST) was obtained through the 7 allele numbers [38,39]. Primers and reaction conditions are shown in S1 Table. Clustered analysis of STs was analyzed through online website (https://www.phyloviz.net/).

## 2.6 Association analysis between MGEs and ESBL genes or VAGs

Correlation coefficient (r) analysis between MGEs and ESBL genes or VAGs were performed using R-software (version 4.2.2) [40]; *P-value* < 0.05 was considered to be statistically significant.

## 2.7 Ethics approval

This study was approved by the Institutional Animal Care and Use Committee of Sichuan Agricultural University under permit number DYY-2018203039.

## 3. Results

### 3.1 Identification of ESBL-EC and distribution of ESBL genes

A total of 69 ESBL-EC (25 strains from Guiyang, 23 strains from Hangzhou, 12 strains from Chongqing, 8 strains from Dalian, 1 strain from Changsha) were isolated from 444 MDR *E. coli*. As shown in Fig 1 and S2 Table, the detection rates of $bla_{CTX-M}$ and $bla_{TEM}$ among the 69 ESBL-EC isolates were 97.10% (67/69) and 52.17% (36/69), respectively. No $bla_{OXA}$ and $bla_{SHV}$ was detected. Moreover, we found 34 ESBL-EC (49.28%, 34/69) carried both $bla_{CTX-M}$ and $bla_{TEM}$. Furthermore, all the ESBL genes were sequenced to identify the variant types of $bla_{CTX-M}$ and $bla_{TEM}$. Six variants of $bla_{CTX-M}$ were identified, including $bla_{CTX-M-55}$ (49.28%, 34/69), $bla_{CTX-M-15}$ (39.13%, 27/69), $bla_{CTX-M-65}$ (4.35%, 3/69), $bla_{CTX-M-14}$ (4.35%, 3/69) and $bla_{CTX-M-3}$ (1.45%,1/69). Three variants of $bla_{TEM}$ were identified, with $bla_{TEM-1}$ (encoding penicillinase) being the most prevalent (28.99%, 20/69), followed by $bla_{TEM-135}$ (encoding penicillinase) (10.14%, 7/69), and $bla_{TEM-198}$ (encoding ESBL) (15.94%, 11/69). Among the five zoos, the most prevalent gene combination pattern in Guiyang Zoo was $bla_{CTX-M-15}$ (32.00%, 8/25), in Hangzhou Zoo, it was $bla_{CTX-M-55}$ & $bla_{TEM-198}$ (34.78%, 8/23), in Chongqing Zoo, it was $bla_{CTX-M-15}$ (25.00%, 3/12), in Dalian Zoo, it was $bla_{CTX-M-15}$ (37.50%, 3/8), and in Changsha Zoo, it was $bla_{CTX-M-55}$ (100.00%, 1/1).

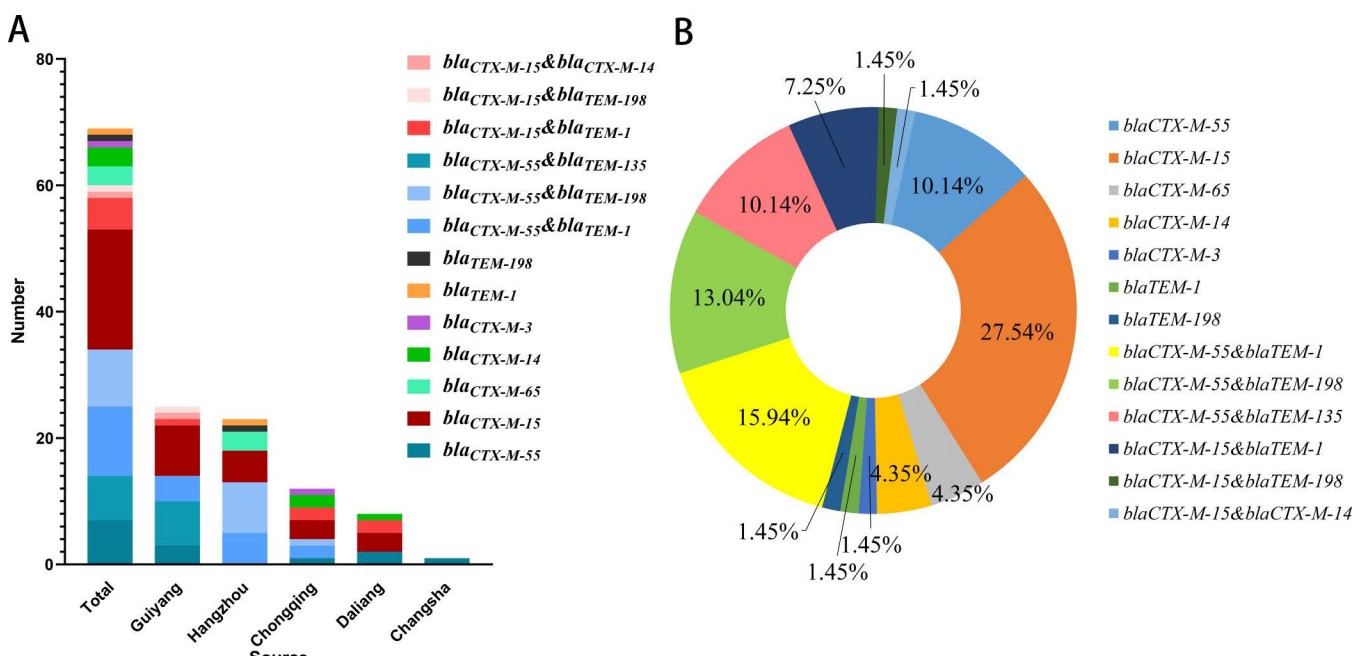

**Fig 1. Distribution of ESBL genes in 69 ESBL-EC strains from captive primates. (A)** Subtypes of ESBL genes carried by 69 ESBL-EC strains; **(B)** Combination patterns of ESBL genes identified from 69 ESBL-EC strains.

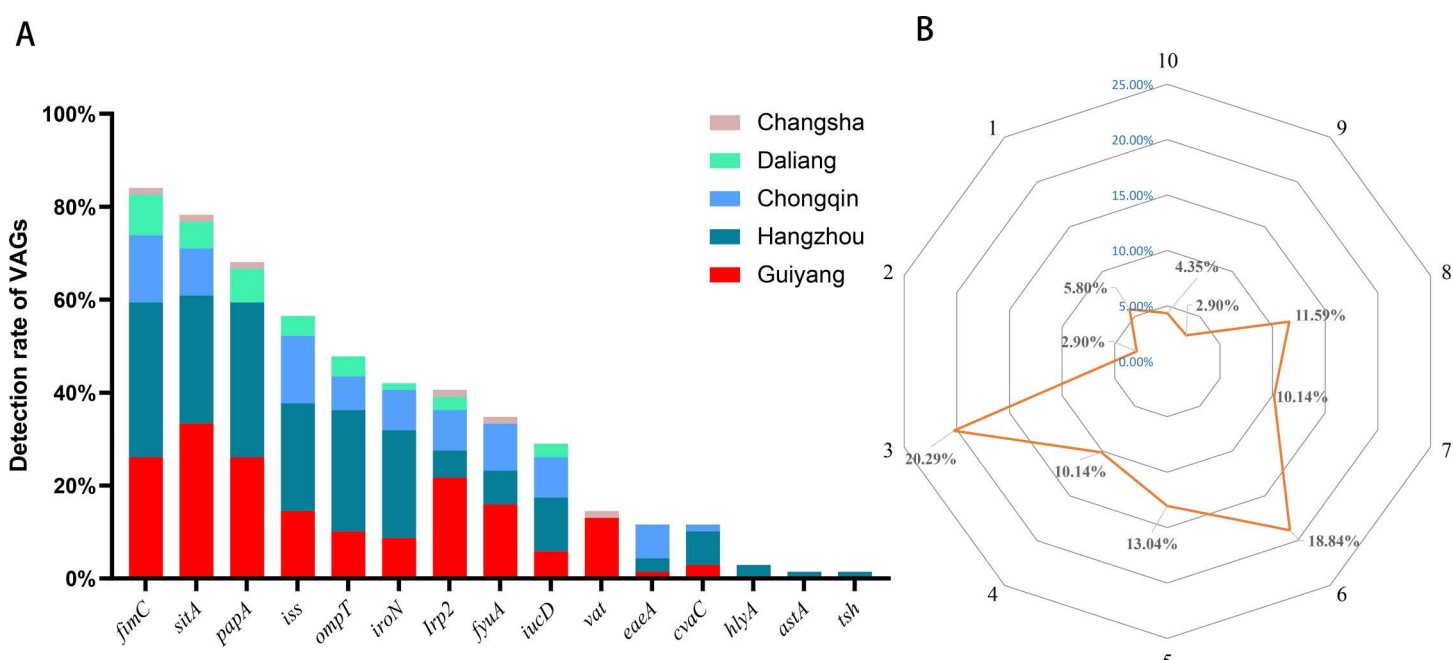

**Fig 2. Distribution of VAGs in 69 ESBL-EC strains from captive primates. (A)** Detection rate of VAGs, and the different colors in each bar represent five zoos; **(B)** The radar map shows the percentages of 69 ESBL-EC strains carrying one to ten types of VAGs. The highest percentage is 20.29% (carrying three types), followed by 18.84% (carrying six types) and the lowest percentage is 2.9% (carrying two or nine types).

**Table 1. VAGs detected from ESBL-EC isolates derived from captive primates.**

| Category of virulence | VAGs | Number of ESBL-EC strains (%) |
|---|---|---|
| Adhesion-related | *fimC* | 58 (84.06) |
| | *papA* | 47 (68.12) |
| | *eaeA* | 8 (11.59) |
| | *tsh* | 1 (1.45) |
| Iron transport related | *sitA* | 54 (78.26) |
| | *iroN* | 29 (42.03) |
| | *Irp2* | 28 (40.58) |
| | *fyuA* | 24 (34.78) |
| | *iucD* | 20 (28.99) |
| Antiserum survival related | *iss* | 39 (56.52) |
| | *ompT* | 33 (47.83) |
| | *cvaC* | 8 (11.59) |
| Invasion and toxin related | *vat* | 10 (14.49) |
| | *astA* | 1 (1.45) |
| Hemolysin related | *hlyA* | 2 (2.90) |

### 3.2 Distribution of VAGs in ESBL-EC isolates

A total of 15 VAGs in 5 categories were detected (*estA*, *estB*, *ipaH*, *elt* and *hlyF* were not detected). As shown in Fig 2 and Table 1, the most prevalent VAGs was *fimC* (84.06%, 58/69), followed by *sitA* (78.26%, 54/69), *papA* (68.12%, 58/69) and *iss* (56.52%, 39/69). The remaining VAGs exhibited detection rates that ranged from 1.45% (*tsh*) to 47.83% (*ompT*). Furthermore, all strains carried at least one VAG. In addition, 91.30% of ESBL-EC isolates carried three or more VAGs. The proportion of strains carrying three VAGs was the most prevalent (20.29%, 14/69), followed by strains carrying six (6/69 18.84%) and five (5/69 13.04%) VAGs. Interestingly, 3 strains (4.35%) harbored ten VAGs (Fig 2B).

### 3.3 Distribution of MGEs in ESBL-EC isolates

A total of ten MGEs were detected in our present study: *IS26* (78.26%, 54/69), *ISEcp1* (60.87%, 42/69), *trbC* (36.23%, 25/69), *tnpA/Tn21* (14.49%, 10/69), *tnsA* (15.94%, 11/69), *merA* (14.49%, 10/69), *ISCR3/14* (10.14%, 7/69), *ISaba1* (7.25%, 5/69), *IS1133* (5.80%, 4/69), and *IS903* (2.90%, 2/69)(S2 Table). The remaining eight MGEs (*traA, traF, tnpU, tnp513, ISkpn7, ISpa7, ISkpn6,* and *ISCR1*) were not detected. Further analysis of the MGEs combinations pattern revealed the presence of 18 distinct patterns (Fig 3), with *IS26 + ISEcp1* being the most prevalent (13/69, 18.84%). A total of 97.10% (67/69) ESBL-EC isolates exhibited the presence of at least one MGE, except for ESBLs-*E. coli* 55 and ESBLs-*E. coli* 56.

### 3.4 Association between MGEs and ESBL genes or VAGs

As shown in Fig 4A, eleven pairs of positive correlations between MGEs and ESBL genes were detected (blaCTX-M-15/ISEcp1, blaCTX-M-15/tnsA, blaCTX-M-65/tnpA/Tn21, blaCTX-M-65/IS1133, blaCTX-M-55/trbC, blaCTX-M-55/merA, blaCTX-M-55/IS26, blaCTX-M-55/ISCR3/14, blaTEM-1/trbC, blaTEM-1/IS26, blaTEM-135/ISCR3/4). Seven pairs of negative correlations were detected (blaCTX-M-15/trbC, blaCTX-M-15/merA, blaCTX-M-15/tnpA/Tn21, blaCTX-M-15/IS26, blaTEM-135/ISEcp1, blaCTX-M-55/ISEcp1, blaCTX-M-55/tnsA). For the association between MGEs and VAGs, a total of 8 pairs

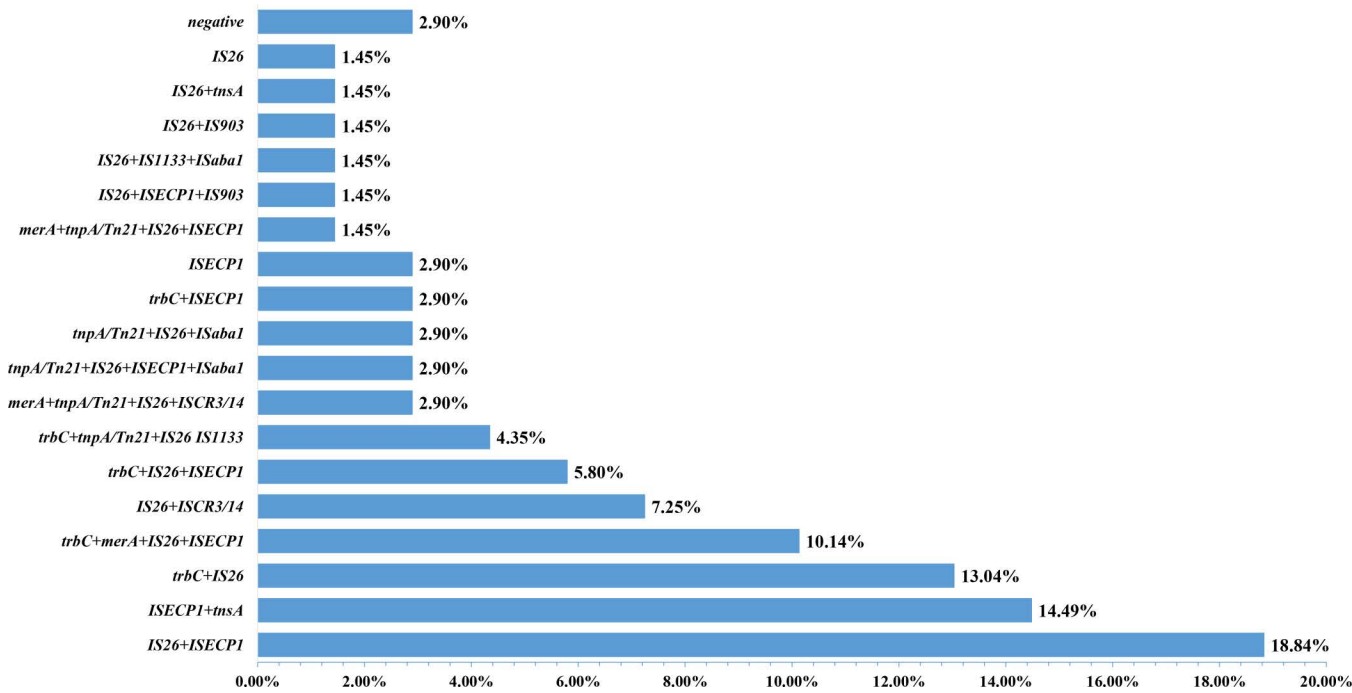

**Fig 3. Combination patterns of MGEs in 69 ESBL-EC isolates from captive primates.**

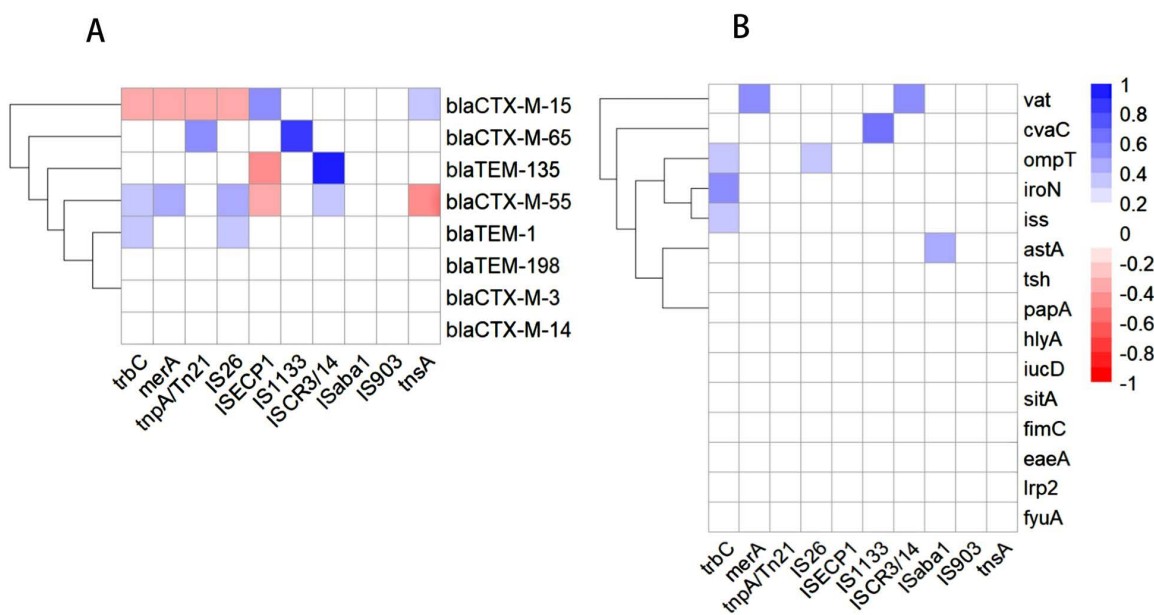

**Fig 4. Heatmap of the association between MGEs and ESBL genes or VAGs in 69 ESBL-EC strains.** Blue represents positive association (r > 0, *P* < 0.05) and red represents negative association (r < 0, *P* < 0.05). The saturation of red and blue represents the r-value. (**A**) Heatmap of the association between ESBL genes and MGEs; (**B**) Heatmap of the association between VAGs and MGEs.

correlations were observed and all the 8 pairs showed positive association (trbC/iss, trbC/iroN, trbC/ompT, IS26/ompT, IS1133/cvaC, merA/vat, ISCR3/14/vat, ISaba1/astA) (Fig 4B).

### 3.5  Conjugative transfer and plasmid incompatibility groups

Our results showed that 28 of 69 ESBL-EC isolates were successfully transferred to the recipient strain J53, and all transconjugants exhibited the ESBL phenotype and were positive for ESBL genes ($bla_{CTX-M}$). The conjugational transfer frequency of the 28 ESBL-EC strains ranging from $1.18 \times 10^{-8}$ to $2.14 \times 10^{-4}$ transconjugants/recipient strain. Twenty-eight transconjugants were found to carry the blaCTX–M gene. ARGs of blaCTX–M, blaTEM, cmlA, floR, qnrS, sul2 and tetA, VAGs of papA, fimC, fyuA, iroN, Irp2, sitA, cvaC, ompT and iss, MGEs of ISEcp1, IS26, trbC, merA, and ISCR3/14 were detected in the transconjugants showed in Fig 5.

The analysis of plasmid incompatibility groups revealed the presence of seven distinct replicons in 28 ESBL-EC isolates, with the most prevalent being IncFIB (78.57%, 22/28), followed by IncFrepB (53.57%, 15/28), IncI1 (14.29%, 4/28), IncFIA (7.14%, 2/28), IncFIC (3.57%, 1/28), IncHI1 (3.57%, 1/12) and IncP (3.57%, 1/28). PBRT confirmed that all 7 replicons were detected in transconjugants (Fig 5).

### 3.6  MLST and Phylogenetic Grouping of 28 ESBL-EC isolates with conjugative transfer ability

The results of phylogenetic grouping showed that among the 28 ESBL-EC isolates with conjugative transfer ability, group B1 (46.43%, 13/28) was the most prevalent phylogroup, followed by group D (25%, 7/28), A (14.28%, 4/28) and B2 (14.28%, 4/28) (Fig 6). MLST analysis showed a high diversity of STs (sequence type), and 11 different STs were observed: ST2161 (10/28, 35.71%), followed by ST442 (7/28, 25.00%), ST2854 (2/28, 7.14%) and ST683 (2/28, 7.14%). The remaining 6 STs (ST69, ST469, ST553, ST155, ST4358 and ST4040) were represented by a single strain each (Fig 6 and S3 Table). Moreover, one ST (strain ESBLs-21) not match the existed STs in the MLST database and was suspected to be new ST, designated as nST1 (S3 Table).

## 4.  Discussion

β-lactams are the most prescribed antibiotics worldwide. However, β-lactams antibiotics are ineffective against ESBLs-producing *Enterobacteriaceae* [41]. ESBL-EC strains were widely detected in captive and wild animals [12–16]. In our results, we successful identified 69 ESBL-EC isolates (15.54%, 69/444), the prevalence was lower than that from captive primates (32.00%) [17] and captive wild animals (24.73%) [19], but higher than reported in another study of primates (7.07%) [20]. Subsequently, the variants of ESBL genes were analyzed in the 69 ESBL-EC strains. Only two categories of ESBL genes ($bla_{CTX-M}$ and $bla_{TEM}$) were detected, with a high diversity observed among $bla_{CTX-M}$ variants. The most prevalent variant was $bla_{CTX-M-55}$ (49.28%, 34/69), followed by $bla_{CTX-M-15}$ (37.68%, 26/69). In our study, the prevalence of $bla_{CTX-M-55}$ was also higher than that of other detected $bla_{CTX-M}$ variants, including $bla_{CTX-M-65}$, $bla_{CTX-M-14}$, and $bla_{CTX-M-3}$. $bla_{CTX-M-55}$ is a variant of $bla_{CTX-M-15}$, which has conferred greater ability to hydrolyze ceftazidime [42,43]. Currently, CTX-M-55 is the most relevant CTX-M enzyme in China. It is the most frequently reported enzyme in animals, and is reported to have an increasing trend in human clinical isolates [44]. $bla_{CTX-M-55}$ is being increasingly detected in a variety of animals, including waterfowl (48%) [35] and poultry (46%) [45]. In addition, $bla_{CTX-M-55}$ has been detected in ESBL-EC from captive primates (37.5%) [19] and has been widely

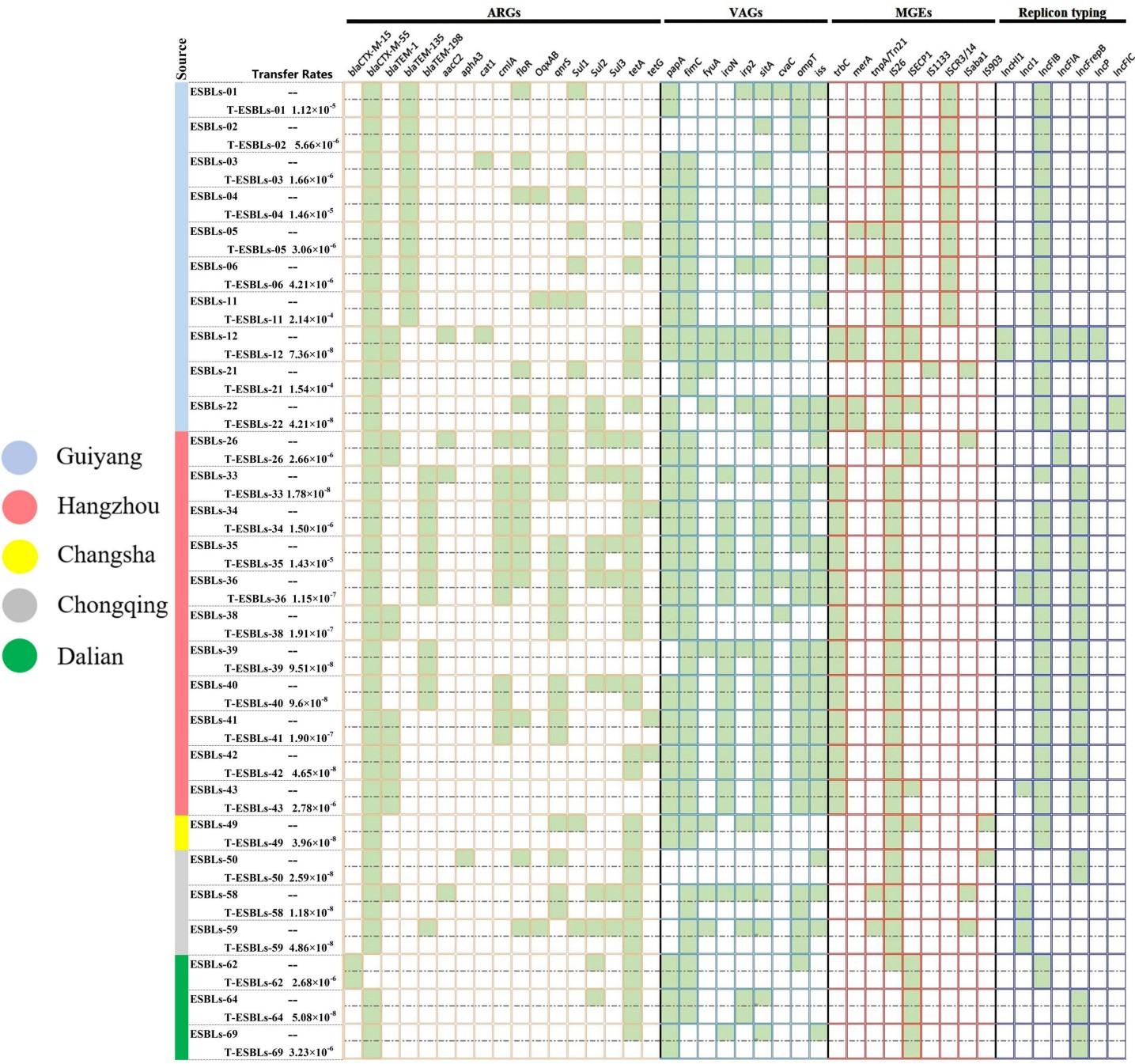

**Fig 5. A heatmap comparing ARGs, VAGs, MGEs, and plasmid replicon types among 28 *E. coli* donor strains and their corresponding transconjugants.** "T" represents that the strain is a transconjugant. Green squares represent donor bacteria or transconjugants carrying the gene or plasmid.

detected in ESBL-EC from other captive wild animals, such as giant pandas (75.00%) [46] and swans (35.71%) [19].

MGEs play an important role in the transmission of ESBL-producing capability, which is mediate by the ESBL genes [22,23]. In our present study, a total of 10 MEGs was detected, with the most prevalent being *IS26* (76.81%, 53/69), followed by *ISEcp1* (60.87%, 42/69).

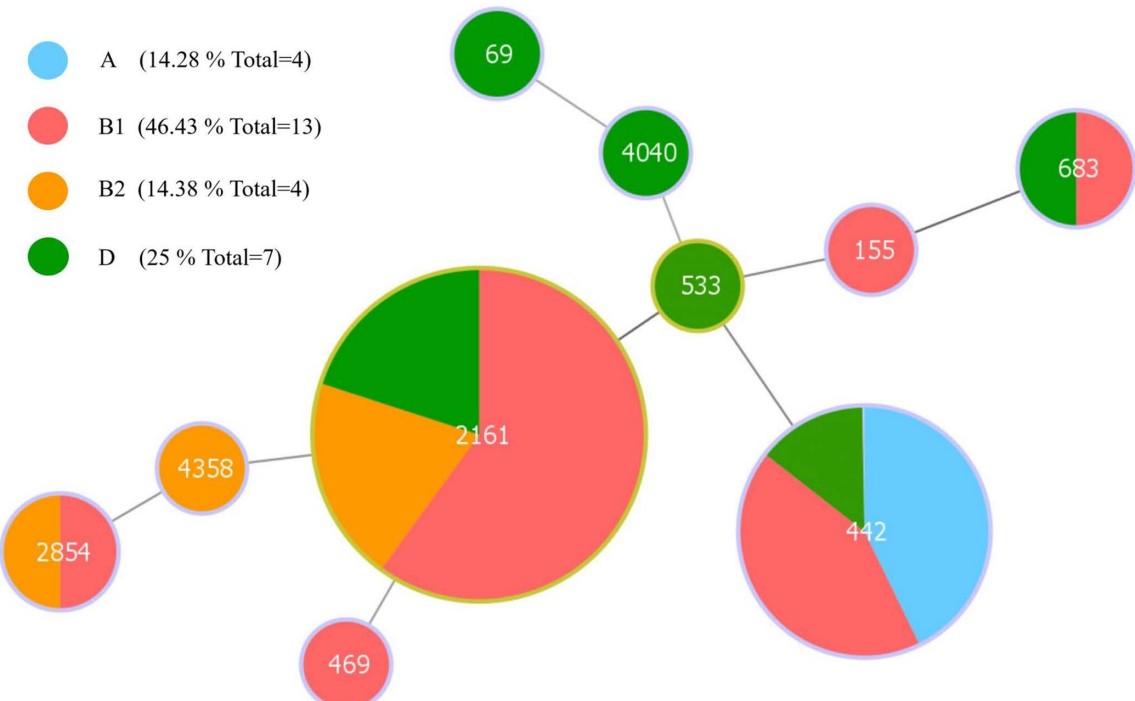

**Fig 6. Minimum spanning tree of MLST types in 28 capable of conjugation ESBL-EC strains (The new ST that did not match in the MLST database was not shown).** The size of the circle is proportional to the number of this ST type. The colors in the circles represent the proportion of the number of phylogroups. The yellow outline of the circle represents the adjacent ST types belonging to the same clonal complex.

These findings are comparable to those reported in waterfowl [47]. A total of 18 combinations between MGEs was detected, with *ISEcp1* + *IS26* combination being the most prevalent (18.84%, n = 13). Our results showed that the ESBL-EC derived from captive primates served as a reservoir for MGEs [48]. *IS26* and *ISEcp1* have been widely reported to be adjacent to $bla_{CTX-M}$ gene and associated with the spread of $bla_{CTX-M}$ [27,28]. *IS903*, which is frequently located the downstream of the ESBL genes, has been reported to be associated with the transmission of the ESBL gene also been detected in our present study [49].

To further evaluate the potential pathogenic risk of ESBL-EC isolates from captive primates, we detected 20 VAGs in 69 ESBL-EC isolates. A total of 15 VAGs were detected and the detection rates ranged from 1.45% to 84.06%. Another study showed 20 out of 24 VAGs were identified in the *E. coli* isolates from clinical healthy captive primates at Como Zoo, with a detection rate ranged from 0.62% to 19.8% [50]. In our study, the most prevalent VAGs was adhesion-related gene *fimC* (84.06%, 58/69), which was not consistent with the report from Como Zoo (*sitA* 19.8%) [50], but similar high prevalence of *fimC* was also detected from giant pandas (91.67%) [51] and dogs (100.00%) [52]. Adhesion-related VAGs are associated with adhesion proteins and bacterial fimbriae, which are responsible for histopathological damage, including adhesion and shedding (attaching-effacing) on the intestinal epithelium, which may result in diarrhea [53,54]. Furthermore, other four categories of VAGs were also detected (including antiserum survival related genes *cvaC, ompT and iss;* hemolysin related genes *hlyA*; invasion and toxin related genes *vat and astA*; iron transport related genes *fyuA, iroN, Irp2 iucD and sitA*). All strains carried at least one VAG and 91.30% of isolates carried three or more VAGs. Previous reports showed that ESBL-EC strains carrying a variety of VAGs have

a high intrinsic virulence potential [31], while another report has demonstrated that community outbreaks of ESBL-EC (CTX-M type) strains have a high rate of VAGs, which contribute to enhancing the fitness of the bacteria [33]. The combination of ESBL-production and the carriage of VAGs in *E. coli* will undoubtedly limit therapeutic options, increase treatment complexity, and enhance pathogenicity [33]. Additionally, strains that carry various VAGs may not be harmful to animals, but there is a potential for them to become pathogenic if host immunity weakens or in response to other adverse environmental factors [21]. The high prevalence of VAGs observed in ESBL-EC isolates from captive primates in our study pose a potential threat to the survival of captive primates in future. To the best of our knowledge, this is the first report to explore the VAGs in ESBL-EC isolates from captive primates.

Furthermore, we analyzed the association of MGEs with ESBL genes or VAGs. We detected 11 positive and 7 negative association pairs between ESBL genes and MEGs, of which the strongest positive correlation was the pair of blaTEM-135 and ISCR3/14. A previous study reported that the ISCR element can move any adjacent DNA sequence without specific recognition [55]. The strong positive correlation observed between ISCR3/14 and blaTEM-135 in our experiments may be related to the trapping of blaTEM-135 by ISCR3/14 [55]. Our study also observed two pairs of positive association (IS26 and blaCTX-M-55; ISEcp1 and blaCTX-M-15) and previous studies showed ISEcp1 and IS26 have been widely reported to be situated upstream of blaCTX-M [56,57]. Negative correlations between ESBL genes and MGEs were also detected. These correlations may be attributed to the presence of specific combinations of ESBL genes with MGEs, but the specific mechanism for these correlations need to be further studied [58,59]. We further analyze the associations between MGEs and VAGs, only positive associations were observed. The TrbC protein plays an important role in conjugative transfer of IncF plasmid [60], and the positive association between trbC and VAGs (iroN, iss or ompT) may be related to the location of VAGs on conjugable plasmids. However, the true relationship between VAGs and ISs is not yet clear [30,61].

Conjugation transfer represents a significant mechanism of HGT [62]. Our results showed that ESBL genes (including $bla_{CTX-M}$ and $bla_{TEM}$) can be transferred horizontally. The plasmid incompatibility groups detected in our study included IncFIB, IncFrepB, IncI1, IncFIA, IncHI1 and IncP. Among them, IncFIB (75%, 21/28) and IncFrepB (53.57%, 15/28) had the higher detection rates. Previous reports showed the IncF plasmids are the epidemic plasmids carrying $bla_{CTX-M-55}$ for transmission, which was also confirmed in our results that $bla_{CTX-55}$ (79.41%, 27/34) had a high conjugation transfer rate [63,64]. Among the 28 ESBL-EC strains with conjugative transfer ability, 96.43% (27/28) were $bla_{CTX-M-55}$-carring isolates, and of these 88.89% (24/27) strains carrying both *IS26* and $bla_{CTX-M-55}$ could be horizontally transferred together. Interestingly, when combined with the results of the associations that revealed a positive association between *IS26* and $bla_{CTX-M-55}$, we speculated that *IS26* is adjacent to $bla_{CTX-M-55}$ in gene structure and can be easily transferred [44,64,65]. Above all, our results showed that MGEs play a significant role in the transmission of ESBL genes. Previous reports showed that the ESBL genes and other ARGs are usually located on the same plasmid that causes MDR and can be horizontally transferred by conjugation [9,10,66]. Our present study showed that *qnrS* (quinolone/fluoroquinolone resistance gene), *tetA* (tetracycline resistance gene), *sul2* (sulfonamides resistance gene), *cmlA* and *floR* (chloramphenicol/phenicol resistance genes) were all detected co-transferred with the plasmid into the recipient strain J53. Moreover, plasmids, particularly IncF plasmids, have been reported as the most common vehicles for the transmission of VAGs, our study also demonstrated that VAGs can be transferred into the recipient strain J53 through conjugation with plasmid [29,30]. Plasmid-mediated HGT can facilitate the transfer of VAGs and provide a pathway for the transmission of pathogenicity between bacteria [67]. Moreover, co-transfer of VAGs with the ESBL genes by plasmid increase the

difficulty of controlling the transmission of the ESBL gene and VAGs. ESBL-EC isolates may pose a serious threat not only to captive primates but also to other animals, humans, and the environment through cloning and direct contact.

Among 28 ESBL-EC strains capable of conjugation transfer, groups A and B1 together accounted for 64.28% (A: 14.28%, B1: 50%). Groups A and B1 are considered to be commensal groups [37] and previous studies on ESBL-EC strains isolated from pigs and dogs also showed that the major ESBL-EC strains belong to commensal groups [68–70]. We also detected groups B2 (14.28%) and D (21.41%), which are classified as potentially pathogenic (37), suggesting that ESBL-EC strains which containing more VAGs may pose a health threat to captive primates. In addition, our result showed that 28 conjugative ESBL-EC isolates exhibited a high population diversity of STs (including 10 known STs and a novel STs.). Of the 10 known STs, ST155 and ST69 are the high-risk clones that have been reported [71], and ST69 has been reported as an important ExPEC carrier and is associated with most extraintestinal diseases [72]. Moreover, ST533 was previously detected in canine ESBL-EC isolates (harboring the $bla_{CTX-M-15}$ gene) [73], ST4358 was identified in *E. coli* isolated from goose feces from China and carried $bla_{CTX-M-55}$ gene [74], ST683 isolate was previously reported in human carbapenem-resistant *E. coli*, which was resistant to both cefotaxime and ceftazidime [75]. ST442 has been reported to be associated with Shiga toxin-producing *E. coli* [76], and ST2161 has been reported in fosfomycin-resistant *E. coli* from humans [77]. As far as we know, ST469, ST2854, and ST4040 have not been reported in ESBL-EC. Our results indicated that ESBL-EC isolates from captive primates exhibited diverse clonal lineages, which may be attributed to cross-infections among different animal species, as well as between humans and captive primates.

## 5. Conclusion and limitations

Our results showed that ESBL-EC strains isolated from captive primates, primarily mediated by $bla_{CTX-M-55}$, were the reservoir of VAGs and MGEs, which pose a potential threat to captive primates, other animals, the environment and humans. Conjugative transfer experiments demonstrated that ESBL genes, ARGs, VAGs and MGEs could be transferred horizontally, which was facilitated by plasmids in highly diverse STs *E. coli*. Based on the One Health concept and the protection of captive primates, we recommend long-term monitoring of the HGT of ESBL-EC isolates from captive primates. Moreover, the use of High-throughput Sequencing or Whole Genome Sequencing could provide more insights into the mechanism of HGT on ESBL-EC strains from captive primates.

## Supporting information

**S1 Table.  PCR primer and conditions used in this study.**
(XLSX)

**S2 Table.  Details of ESBL genes, VAGs and MGEs in 69 ESBL-EC strains from captive primates.**
(XLSX)

**S3 Table.  Distribution of sequence types (STs) and phylogenetic grouping in28 ESBL-EC strains isolates with conjugative transfer ability.**
(XLSX)

## Acknowledgement

We thank Dr Junai Gan and Lei Deng for the English language revision.

## Author contributions

**Conceptualization:** Keyun Shi, Zhijun Zhong.

**Data curation:** Wenhao Zhong, Yuxin Zhou, Mengjie Che.

**Investigation:** Liqin Wang, Xingyu Tian.

**Methodology:** Keyun Shi, Zhijun Zhong.

**Software:** Chengdong Wang, Yuehong Cheng, Haifeng Liu, Ziyao Zhou, Guangneng Peng, Kun Zhang, Yan Luo, Keyun Shi, Zhijun Zhong.

**Validation:** Chengdong Wang, Yuehong Cheng, Haifeng Liu, Ziyao Zhou, Guangneng Peng, Kun Zhang, Yan Luo.

**Visualization:** Wenhao Zhong, Yuxin Zhou, Mengjie Che.

**Writing – original draft:** Wenhao Zhong, Yuxin Zhou, Mengjie Che.

**Writing – review & editing:** Wenhao Zhong, Zhijun Zhong.

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
