## [Decision Letter · Decision Letter 0]

19 Jan 2025

PONE-D-24-55572Extended-spectrum β-lactamase-producing Escherichia coli isolated from captive primates: characteristics and horizontal gene transfer ability analysisPLOS ONE

Dear Dr. Zhong,

Thank you for submitting your manuscript to PLOS ONE. After careful consideration, we feel that it has merit but does not fully meet PLOS ONE’s publication criteria as it currently stands. Therefore, we invite you to submit a revised version of the manuscript that addresses the points raised during the review process.

We look forward to receiving your revised manuscript.

Kind regards,

Gabriel Trueba, PhD

Academic Editor

PLOS ONE

“This work was funded by Study on Key Technologies for Conservation of Wild Giant Panda Populations and Its Habitats within Giant Panda National Park System (CGF2024001), and Chengdu Giant Panda Breeding Research Foundation (CPF2017-05, CPF2015-4).

Reviewers' comments:

Reviewer's Responses to Questions

**Comments to the Author**

1. Is the manuscript technically sound, and do the data support the conclusions?

Reviewer #1: Yes

Reviewer #2: Yes

2. Has the statistical analysis been performed appropriately and rigorously? 

Reviewer #1: Yes

Reviewer #2: Yes

3. Have the authors made all data underlying the findings in their manuscript fully available?

Reviewer #1: Yes

Reviewer #2: Yes

4. Is the manuscript presented in an intelligible fashion and written in standard English?

Reviewer #1: Yes

Reviewer #2: Yes

5. Review Comments to the Author

Reviewer #1: This manuscript describes the characteristics of Escherichia coli isolated from captive primates. The objective is clear, and the methods used are appropriate. While the results offer some value to readers in this specific field, they do not necessarily present novel findings. I provide the following editorial comments and questions to help improve the readability of the manuscript. Additionally, I strongly recommend that the authors have their revised manuscript undergo thorough English editing before resubmission

Line 29: remove E. coli

Line 30: Replace all instances of Escherichia coli with E. coli as an abbreviation throughout the entire manuscript.

Line 33: see comment in line 30.

Line 46: Remove one occurrence of 'captive' as it is repeated.

Line 76: E. coli should be italicized. This is an incomplete sentence.

Line 93: please check irp2 or Irp2 here.

Line 104: used as the recipient strain

Line 105: Those colonies capable….

Line 121: This is an incomplete sentence.

Line 131: Replace the comma (,) with a semicolon (;).

Line 144-148: You might consider rephrasing it as: Among the five zoos, the most prevalent gene combination pattern in Guiyang Zoo was blaCTX-M-15 (32.00%, 8/25); in Hangzhou Zoo, it was blaCTX-M-55 & blaTEM-198 (34.78%, 8/23); in Chongqing Zoo, it was blaCTX-M-15 (25.00%, 3/12); in Dalian Zoo, it was blaCTX-M-15 (37.50%, 3/8); and in Changsha Zoo, it was blaCTX-M-55 (100%, 1/1).

Line 157: replace greatest with largest

Line 205-206: Please explain how ESBLs produced by Enterobacteriaceae are ineffective against β-lactam drugs? Do you mean “β-lactam drugs are ineffective against ESBLs produced by Enterobacteriaceae”?

Line 211: Only two categories of ESBLs-genes…

Line 216: in a variety of animals

Line 246-248: You might consider rephrasing it as: The combination of ESBL production and the carriage of virulence-associated genes (VAGs) in E. coli will undoubtedly limit therapeutic options, increase treatment complexity, and enhance pathogenicity.

Line 248-250: You might consider rephrasing it as: Additionally, strains that carry various VAGs may not be harmful to animals, but there is a potential for them to become pathogenic if host immunity weakens or in response to other adverse environmental factors.

Line 262: The Trbc protein

Line 267-269: You might consider rephrasing it as: The plasmid incompatibility groups detected in our study included IncFIB, IncFrepB, IncI1, IncFIA, IncHI1, and IncP.

Line 283-285: You might consider rephrasing it as: Moreover, plasmids, particularly IncF plasmids, have been reported as the most common vehicles for the transmission of VAGs. Our study demonstrated that 9 VAGs can be transferred into the recipient strain J53 through conjugation with plasmids.

Line 296-300: The ST detected in this study requires a more detailed discussion.

Reviewer #2: 1. In the abstract you have to mention the studied samples, their number, their origins, region of the study. The aim not mentioned, the experiment were not mentioned. Perhaps you are limited by the number of words that must in the abstract but few data are needed to understand the work by learning the abstract.

2. Abstract, line 19, write ‘...The population structure analysis showed that the phylogroup B1 and ST2161 were the most prevalent...’

3. Line 20 you wrote ‘ESBL-EC derived from captive primates pose a potential threat to animals, the environment, and humans through horizontal gene transfer.’ In reality the threat is by direct contact with colonized primates or by environmental pollution, so horizontal transfer is not a strict threat, perhaps genetic transfer is more implicated in the transfer between several genera colonizing primate. Take this in mind and try to modify this sentence in the abstract and this idea in the remaining part of your manuscript.

4. In general we say ‘Mobile genetic element(s) (MGE)’ not ‘Mobile gene element (MGE)’ so modify this in the keywords and in the manuscript.

5. Write ‘Keywords: Captive primate; Escherichia coli; Extended‐spectrum β‐lactamase (ESBL), Virulence-associated gene (VAG); Mobile genetic elements (MGEs); Horizontal gene transfer (HGT).

6. Line 30, write ‘...Antimicrobial resistant (AMR) E. coli poses a threat to animal............ The production of extended-spectrum β-lactamases (ESBLs) by E. coli represents the primary mechanism responsible for the treatment failure by β-lactam antibiotics (5). The ESBL-producing E. coli (ESBL-EC) is one of the most important pathogens at the One...... CTX-Ms are the most predominant types of ESBLs which are encoded by several blaCTX-M type genes (7).’

7. In all the manuscript including the abstract the name of bacteria must be in italic

8. Please verify the list of references, you need not to write words with capital letters, see the guideline of the journal and a recent article published in the journal. For example change ‘Negeri AA, Mamo H, Gurung JM, Firoj Mahmud AKM, Fällman M, Seyoum ET, et al. Antimicrobial Resistance Profiling and Molecular Epidemiological Analysis of Extended Spectrum β-Lactamases Produced by Extraintestinal Invasive Escherichia coli Isolates From Ethiopia: The Presence of International High-Risk Clones ST131 and ST410 Revealed. Front Microbiol. 2021;12:706846.’ BY ‘‘Negeri AA, Mamo H, Gurung JM, Firoj Mahmud AKM, Fällman M, Seyoum ET, et al. Antimicrobial resistance profiling and molecular epidemiological analysis of extended spectrum β-lactamases produced by extraintestinal invasive Escherichia coli isolates from Ethiopia: The presence of international high-risk clones ST131 and ST410 revealed. Front Microbiol. 2021;12:706846.’ VERIFY ALL REFERENCES.

9. In this sentence ‘ESBL-EC has been frequently reported in captive and wild animals, such as giant pandas, ostriches 41 and white storks (12-15).’ YOU CAN ADD THIS REFERENCE AS REVIEW ARTICLE: Abbassi MS, Badi S, Lengliz S, Mansouri R, Salah H, Hynds P. Hiding in plain sight-wildlife as a neglected reservoir and pathway for the spread of antimicrobial resistance: a narrative review. FEMS Microbiol Ecol. 2022;98:fiac045.

10. Line 44, write ‘In 2022, Zeng et al (18) isolated eight ESBL-EC strains from captive primates, all of which carried the blaCTX-M gene. In 2024, Bazalar-Gonzales et al (19) isolated seven ESBL-EC strains from captive and semi-captive primates, three blaCTM-X variants (blaCTX-M-15, blaCTX-M-55 and blaCTX-M-65) were identified.

11. Line 57, write ‘Insertion sequences (ISs) have been frequently reported to be associated with the transfer of ESBLs genes, for example, ISEcp1 and IS26 have......... Because of this, blaCTX-M has rapidly become the most common ESBLs gene transferred by MGEs (26, 27)............. enhanced fitness of these emerging CTX-M-harboring strains (30, 31).’

12. Write ‘2.3 Screening of ESBL genes, VAGs and MGEs from ESBL-EC isolates’.

13. The occurrence of the following mobile MGEs were investigated: ISEcp1, IS26, IS903, ISCR1, ISCR3/14, traA, trbC, traF, tnpU, tnpA/Tn21, tnsA, merA, tnp513, IS1133, ISpa7, ISaba1, ISkpn6, ISkpn7......... for the PCR primers were presented in Table S1.’

14. PLEASE PROVIDE the reference of the conjugation experiment.

15. Write ‘’In this study, the sodium azide- resistant E. coli J53 was used...... Those capable of growing on Luria-Bertani (LB) agar supplemented with CTX (4 mg/mL) and sodium azide (150 mg/mL)..’

16. Please this sentence is not clear, improve it ‘The dilution factor (30 to 300 single colonies on solid medium) was determined experimentally.’

17. Write ‘.....sulfonamides (sul1, sul2, sul3), quinolones/fluoroquinolones (qnrS, oqxAB), chlormphenicol/phenicol (flor, cmlA, cat1) … We then examined whether the transconjugants carry these same genes. In addition, VAGs,..... recognize K, B/O, FrepB, FIB, W, I1, FIA, FIIA, N, T, HI2, X, Y, FIC, A/C, L/M, P and HI1. Primers and reaction conditions were listed in table S1.’

18. Write ‘Briefly, seven housekeeping genes（purA, recA, gyrB, icd, mdh, adk and fumC） were investigated by PCR, the obtained DNA fragments were sequenced and uploaded to the website (PLEASE INDICATE THE NAME OF THE WEBSITE) to obtain the allele number, and the sequence type (ST)......... Primers and reaction conditions are shown in table S1’.

19. Write ‘..As shown in figure 1 and table S2, the detection........

20. Please in line 143-144, indicate if these TEM variants are ESBLor penicillinase; if you prefer to say this in the discussion no problem, but you should say this..

21. ‘As shown in Figure 2 and table 1, the most prevalent VAGs was fimC.....’

22. Write ‘ The proportion of strains carrying three VAGs was the most prevalent (20.29 %, 14/69), followed by strains carrying six (6/69; 18.84 % ) and five (5/69; 13.04%) VAGs.

23. I changed this sentence ‘The strain carrying the greatest number of virulence genes was 10, accounting 158 for 4.35 % (3/69) of the total (Fig 2B).’ By ‘Interestingly, 3 strains (4.35%) harbored ten VRGs (Figure 2B).’ THE SENTENCE WAS NOT CLEAR, PLEASE VERIFY AND MAKE CORRECTION.

24. In material and method you mentioned that you searched 18 MGEs including ISEcp1, IS26, IS903, ISCR1, ISCR3/14, traA, trbC, traF, tnpU, tnpA/Tn21, tnsA, merA, tnp513, IS1133, ISpa7, ISaba1, ISkpn6, ISkpn7. But in the list of primers there are several primers missed for example those of ISkpn6, ISkpn7, traA, .... PLEASE VERIFY and add them in the table of primers. PLEASE VERIFY ALSO THIS IN THE SUPPLEMENTARY file, TABLE S1.

25. Write ‘...with IS26+ISEcp1 being the most prevalent (13/69; 18.84 %). ......... MGE, except for ESBLs-E.coli 55 and ESBLs-E.c56.’

26. Line 174, please verify, there some combinations are repeated twice, for example blaCTX-M-65/tnpA/Tn21, blaCTX-M-65/IS1133. I modifiy as follows: ‘(blaCTX-M-15/ISEcp1, blaCTX-M-15/tnsA, blaCTX-M-55/trbC, blaCTX-M-55/merA, blaCTX-M-55/IS26, blaCTX-M-55/ISCR3/14, blaCTX-M-65/tnpA/Tn21, blaCTX-M-65/IS1133, blaCTX-M-55/trbC, , blaCTX-M-65/tnpA/Tnp21, blaCTX-M-65/IS1133, blaTEM-1/trbC, blaTEM-1/ISEcp1)’. ALSO VERIFY THESE COMBINATION: ‘Seven 175 pairs of negative correlations were detected (blaCTX-M-15/trbC, blaCTX-M-15/merA, blaCTX-M176 15/tnpA/Tn21, blaCTX-M-15/IS26, blaTEM-135/ISEcp1, blaCTX-M-15/trbC, blaCTX-M-15/merA, blaTEM-M177 15/tnpA/Tn21, blaCTX-M-15/IS26, blaTEM-135/ISEcp1, blaCTX-M-55/ISEcp1, blaCTX-M-55/tnsA).’ Verify and correct.

27. What mean ‘negative correlations’ please explain it by single sentence in this sentence:’.. Seven pairs of negative correlations were detected (blaCTX-M-15/trbC..’

28. Line 182, you wrote ‘Our result showed that 28 of 69 ESBL-EC isolates were successfully transferred to the recipient strain J53’ PLEASE what was transferred? Cefotaxime resistance? ESBL phenotype?. Improve the sentence.

29. Write ‘The conjugational transfer frequency of the 28 ESBL-EC strains ranging from 1.18×10- 8 to 2.14×10-4 tranconjugants/recipient strain.’

30. Line 197, write ‘..MLST analysis showed a high diversity of STs (sequence type), and 11 different STs were observed: ST2161 (10/28; 35.71 %), followed by ST442 (7/28; 25.00 %), ST2854 200 (2/28; 7.14 %) and ST683 (2/28; 7.14 %).’

31. Please in table S1, change Amide alcohols by ‘chlormphenicol/phenicol’

32. In discussion line 205, the following sentence is not correct, you mean thing but you write other thing ‘However, ESBLs produced by Enterobacteriaceae are ineffective against β-lactam drugs (39).’ I propose :’ β-lactams are the most prescribed antibiotics worldwide. However, β-lactam antibiotics are ineffective against ESBLs-producing Enterobacteriaceae (39).’

33. Line 216, improve this sentence, it is not clear ‘The prevalence of the blaCTX-M-55 in our present study is 214 higher than that of another variant of blaCTX-M’.

34. Line 219, please highlight that the CTX-M-55 is the most relevant CTX-M enzyme in China (please verify if it is relevant from animal and human isolates), I believe also in other Asian countries, in other European countries and Africa CTX-M-1 is relevant in animal but CTX-M-15 IN HUMAN ISOLATES. This is very important to say that the scenario in animal mirror the scenario (the epidemiology of ESBL enzymes) in human isolates. A large view of the epidemiology of ESBL enzymes is requested to understand the ways of ESBL transmission in Humn/Animal/environmental interface under the One Health approach.

35. Write ‘MGEs play an important role in the transmission of ESBL-producing capability, which is mediate by the ESBL genes (21, 22)’

36. In all your manuscript you have to write ‘ESBL genes’ not ‘ESBLs genes’, also not ‘ESBL genes’

37. Xorrect this sentence by ‘Our present study showed that qnrS (quinolone/fluoroquinolone resistance gene), tetA (tetracycline resistance gene), sul2 (sulfonamides resistance gene), cmlA and floR (chlormphenicol/phenicol resistance genes)’

38. Line 285, you performed PCR of virulence gene in the recipient strain?, I believe that this strain harbor papA and fimC; if you did not so you cannot say ‘Notably, all papA and fimC were detected in the corresponding transconjugants.’ Since that two gene can already exist in the recipient strain especially when taken in mind that these genes encode simple fimbrae, very common in E. coli isolates. Verify this point.

39.

6. PLOS authors have the option to publish the peer review history of their article (what does this mean? ). If published, this will include your full peer review and any attached files.

**Do you want your identity to be public for this peer review?** For information about this choice, including consent withdrawal, please see our Privacy Policy .

Reviewer #1: No

Reviewer #2: No

---

## [Author Response · Author response to Decision Letter 1]

1 Feb 2025

Dear Editor:

Thank you very much for your letter and the reviewers’ comments regarding our manuscript submitted to “PLOS ONE” (Manuscript ID: PONE-D-24-55572). We have checked the article and revised it according to the comments, and carefully proof-read the manuscript. The language presentation was improved with assistance from English speakers with appropriate research background. All revisions were marked in revised manuscript with track changes.

We hope, with these modifications and improvements based on your suggestion and the reviewer’s comments, the quality of our manuscript would meet the publication standard of “PLOS ONE”. Once again, we acknowledge your comments and constructive suggestions very much, which are valuable in improving the quality of our manuscript.

We submit here the revised manuscript with track changes as well as a list of changes. Thanks again for your reconsideration of our manuscript for publication in your journal. If you have any question about this paper, please don’t hesitate to let me know.

Best wishes for you!

Sincerely yours, Zhijun Zhong

College of Veterinary Medicine

Key Laboratory of Animal Disease and Human Health of Sichuan Province

Sichuan Agricultural University

Chengdu 611130, P. R. China

E-mail: zhongzhijun488@126.com

Response to Reviewer 1 Comments

Response to Reviewer:

Thank you for reading the manuscript carefully and providing many thoughtful comments, which will significantly improve the quality of the manuscript. We have carefully revised the manuscript in accordance with your comments (Tracked Version). Detailed responses are as follows:

Reviewer #1: This manuscript describes the characteristics of Escherichia coli isolated from captive primates. The objective is clear, and the methods used are appropriate. While the results offer some value to readers in this specific field, they do not necessarily present novel findings. I provide the following editorial comments and questions to help improve the readability of the manuscript. Additionally, I strongly recommend that the authors have their revised manuscript undergo thorough English editing before resubmission

Thank you for your suggestions about the language of the manuscript, we carefully checked the English expression and made English revisions under the scrutiny of native English speakers with appropriate research background.

Comment 1: Line 29: remove E. coli

Response 1: Thanks for your comments, we have removed E. coli in our revised manuscript (Escherichia coli is a resident microorganism in the gut of humans and…) (Tracked Version Line 40).

Comment 2: Line 30: Replace all instances of Escherichia coli with E. coli as an abbreviation throughout the entire manuscript.

Response 2: Thanks for your comments, we have checked the abbreviation error and corrected Escherichia coli to E. coli in our revised manuscript.

Comment 3: Line 33: see comment in line 30.

Response 3: Thanks for your comments, we have corrected the error in our revised manuscript (The ESBL-producing E. coli (ESBL-EC) is one of the most important pathogens at the One Health interface…) (Tracked Version Line 45).

Comment 4: Line 46: Remove one occurrence of 'captive' as it is repeated.

Response 4: Thanks for your comments, we have removed the repeated 'captive' in our revised manuscript (In 2024, Bazalar-Gonzales et al [20] isolated seven ESBL-EC strains from captive and semi-captive primates, three blaCTM-X variants...) (Tracked Version Line 62).

Comment 5: Line 76: E. coli should be italicized. This is an incomplete sentence.

Response 5: Thanks for your comments, we have corrected the unitalicized error, checked the grammar and revised the sentence to make the sentence structure complete as follow: ‘Four hundred and forty-four MDR E. coli strains from captive primates, which were isolated from 13 zoos and had previously been stored in our laboratory, were used in this study’ (Tracked Version Line 95-97).

Comment 6: Line 93: please check irp2 or Irp2 here.

Response 6: Thanks for your comments. After checking, we confirmed that it was Irp2 and corrected the error in our revised manuscript.

Comment 7: Line 104: used as the recipient strain

Response 7: Thanks for your comments, we have corrected the error in our revised manuscript (In this study, the sodium azide-resistant E. coli J53 was used as the recipient strain to evaluate the conjugation and transfer ability of…) (Tracked Version Line 129).

Comment 8: Line 105: Those colonies capable….

Response 8: Thanks for your comments, we have added the word to make the sentence correct (Those colonies capable of growing on Luria-Bertani (LB) agar supplemented with CTX (4 mg/mL) and sodium aside (100 mg/mL) were…) (Tracked Version Line 130).

Comment 9: Line 121: This is an incomplete sentence.

Response 9: Thanks for your comments, we carefully checked the sentences for grammatical errors and corrected them as follows: ‘Phylogenetic analysis (A, B1, B2, D) was conducted by detecting yjaA, chuA, and TSPE4.C2, according to Clermont et al’ (Tracked Version Line 150-151).

Comment 10: Line 131: Replace the comma (,) with a semicolon (;).

Response 10: Thank you for pointing out the error, we have corrected it in our revised manuscript (…VAGs were performed using R-software (version 4.2.2) [40]; P-value...) (Tracked Version Line 163).

Comment 11: Line 144-148: You might consider rephrasing it as: Among the five zoos, the most prevalent gene combination pattern in Guiyang Zoo was blaCTX-M-15 (32.00%, 8/25); in Hangzhou Zoo, it was blaCTX-M-55 & blaTEM-198 (34.78%, 8/23); in Chongqing Zoo, it was blaCTX-M-15 (25.00%, 3/12); in Dalian Zoo, it was blaCTX-M-15 (37.50%, 3/8); and in Changsha Zoo, it was blaCTX-M-55 (100%, 1/1).

Response 11: Thank you for the detailed suggestions for revision. We have revised the sentence accordingly in our revised manuscript (Among the five Zoos, the most prevalent gene combination pattern in Guiyang Zoo was blaCTX-M-15 (32.00%, 8/25), in Hangzhou Zoo, it was blaCTX-M-55 & blaTEM-198 (34.78%, 8/23), in Chongqing Zoo, it was blaCTX-M-15 (25.00%, 3/12), in Dalian Zoo, it was blaCTX-M-15 (37.50%, 3/8), and in Changsha Zoo, it was blaCTX-M-55 (100.00%, 1/1).) (Tracked Version Line 183-186).

Comment12: Line 157: replace greatest with largest

Response 12: Thanks for your comments. We have revised this sentence in our revised manuscript.

Comment 13: Line 205-206: Please explain how ESBLs produced by Enterobacteriaceae are ineffective against β-lactam drugs? Do you mean “β-lactam drugs are ineffective against ESBLs produced by Enterobacteriaceae”?

Response 13: Thank you for your comment, ESBLs are produced by some bacteria such as Enterobacteriaceae and are able to hydrolyze β-lactam antibiotics, thus leading to bacterial resistance to these antibiotics. Sorry for the wrong expression here, we have corrected the sentence as follows: "β-lactams are the most prescribed antibiotics worldwide. However, β-lactams antibiotics are ineffective against ESBL-producing Enterobacteriaceae." (Tracked Version Line 253)

Comment 14: Line 211: Only two categories of ESBLs-genes…

Response 14: Thank you for your comment, we have corrected it in our revised manuscript (Only two categories of ESBL genes (blaCTX-M and blaTEM) were…) (Tracked Version Line 259).

Comment 15: Line 216: in a variety of animals

Response 15: Thank you for your comment, we have corrected it in our revised manuscript (blaCTX-M-55 is being increasingly detected in a variety of animals, including waterfowl…) (Tracked Version Line 269).

Comment 16: Line 246-248: You might consider rephrasing it as: The combination of ESBL production and the carriage of virulence-associated genes (VAGs) in E. coli will undoubtedly limit therapeutic options, increase treatment complexity, and enhance pathogenicity.

Response 16: Thank you for your comments. Your opinions make the expression more scientific and rigorous. We have revised it according to your suggestions in our revised manuscript (Tracked Version Line 305-307).

Comment17: Line 248-250: You might consider rephrasing it as: Additionally, strains that carry various VAGs may not be harmful to animals, but there is a potential for them to become pathogenic if host immunity weakens or in response to other adverse environmental factors.

Response 17: Thank you for your comments. We have revised the sentence in our revised manuscript (Tracked Version Line 310-312).

Comment 18: Line 262: The Trbc protein

Response 18: Thanks for pointing out the error, we have corrected it in our revised manuscript (The TrbC protein plays an important role in conjugative transfer…) (Tracked Version Line 330).

Comment 19: Line 267-269: You might consider rephrasing it as: The plasmid incompatibility groups detected in our study included IncFIB, IncFrepB, IncI1, IncFIA, IncHI1, and IncP.

Response 19: Thank you for your comments, we have revised the sentence in our revised manuscript (Tracked Version Line 337).

Comment 20: Line 283-285: You might consider rephrasing it as: Moreover, plasmids, particularly IncF plasmids, have been reported as the most common vehicles for the transmission of VAGs. Our study demonstrated that 9 VAGs can be transferred into the recipient strain J53 through conjugation with plasmids.

Response 20: Thank you for your comments. We have improved the sentence according to the suggestions you provided in our revised manuscript (Tracked Version Line 355-358).

Comment 21: Line 296-300: The ST detected in this study requires a more detailed discussion.

Response 21: Thank you for your comments and we have discussed the detected STs in more detail in our revised manuscript as follow: ‘Moreover, ST533 was previously detected in canine ESBL-EC isolates (harboring the blaCTX-M-15 gene) [73], ST4358 was identified in E. coli isolated from goose feces from China and carried blaCTX-M-55 gene [74], ST683 isolate was previously reported in human carbapenem-resistant E. coli, which was resistant to both cefotaxime and ceftazidime [75]. ST442 has been reported to be associated with Shiga toxin-producing E. coli [76], and ST2161 has been reported in fosfomycin-resistant E. coli from humans [77]. As far as we know, ST469, ST2854, and ST4040 have not been reported in ESBL-EC. Our results indicated that ESBL-EC isolates from captive primates exhibited diverse clonal lineages, which may be attributed to cross-infections among different animal species, as well as between humans and captive primates.’ (Tracked Version Line 378-387).

Response to Reviewer 2 Comments

Response to Reviewer:

Thank you for reading the manuscript carefully and providing many thoughtful comments, which will significantly improve the quality of the manuscript. We have carefully revised the manuscript in accordance with your comments (Tracked Version). Detailed responses are as follows:

Reviewer #2:

Comment 1: In the abstract you have to mention the studied samples, their number, their origins, region of the study. The aim not mentioned, the experiment were not mentioned. Perhaps you are limited by the number of words that must in the abstract but few data are needed to understand the work by learning the abstract.

Response 1: Thanks for your comments, we have improved the abstract section. We have added research objectives, sample information such as sources, quantities, and experimental methods to the abstract in our revised manuscript (The rapid spread of extended-spectrum β-lactamases (ESBLs)-producing Escherichia coli (ESBL-EC) around the world has become a significant challenge for humans and animals. In this study, we aimed to examine the characteristics and horizontal gene transfer (HGT) capacity of ESBL-EC derived from captive primates. We screened for ESBL-EC among a total of 444 multidrug-resistant (MDR) E. coli strains isolated from 13 zoos in China using double-disk test. ESBL genes, mobile genetic elements (MGEs), and virulence-associated genes (VAGs) in ESBL-EC were detected through polymerase chain reaction (PCR). Furthermore, conjugation experiments were conducted to examine the HGT capacity of ESBL-EC, and the population structure (phylogenetic groups and MLST) was determined.) (Tracked Version Line 6-12).

Comment 2: Abstract, line 19, write ‘...The population structure analysis showed that the phylogroup B1 and ST2161 were the most prevalent...’

Response 2: Thanks for your comment, we have corrected the sentence in the abstract section in our revised manuscript (Tracked Version Line 26-27).

Comment 3: Line 20 you wrote ‘ESBL-EC derived from captive primates pose a potential threat to animals, the environment, and humans through horizontal gene transfer.’ In reality the threat is by direct contact with colonized primates or by environmental pollution, so horizontal transfer is not a strict threat, perhaps genetic transfer is more implicated in the transfer between several genera colonizing primate. Take this in mind and try to modify this sentence in the abstract and this idea in the remaining part of your manuscript.

Response 3: Thank you for your comments. We have revised the manuscript to make the expression correct and scientific in our revised manuscript. Our revisions are as follows: ESBL-EC poses a potential threat to captive primates and may spread to other animals, humans, and the environment. It is imperative to implement measures to prevent the transmission of ESBL-EC among captive primates (Tracked Version Line28-30); ESBL-EC isolates may pose a serious threat not only to captive primates but also to other animals, humans, and the environment through cloning and direct contact (Tracked Version Line363-365).

Comment 4: In general we say ‘Mobile genetic element(s) (MGE)’ not ‘Mobile gene element (MGE)’ so modify this in the keywords and in the manuscript.

Response 4: Thanks for your comment, we have corrected the problem and made corrections to the errors elsewhere in our revised manuscript.

Comment 5: Write ‘Keywords: Captive primate; Escherichia coli; Extended‐spectrum β‐lactamase (ESBL), Virulence-associated gene (VAG); Mobile genetic elements (MGEs); Horizontal gene transfer (HGT).

Response 5: Thanks for your comments, we have adjusted the order of keywords in our revised manuscript (Keywords: Captive primate; Escherichia coli; Extended‐spectrum β‐lactamase (ESBL); Virulence-associated gene (VAG); Mobile genetic element (MGE); Horizontal gene transfer (HGT)) (Tracked Version Line34-37).

Comment 6: Line 30, write ‘...Antimicrobial resistant (AMR) E. coli poses a threat to animal............ The production of extended-spectrum β-lactamases (ESBLs) by E. coli represents the primary mechanism responsible for the treatment failure by β-lactam antibiotics (5). The ESBL-producing E. coli (ESBL-EC) is one of the most important pathogens at the One...... CTX-Ms are the most predominant types of ESBLs which are encoded by several blaCTX-M type genes (7).’

Response 6: Thank you for your comments, we have checked and corrected the sentence in our revised manuscript (Antimicrobial resistant (AMR) E. coli poses a threat to animal and human health [3, 4]. The production of extended-spectrum β-lactamases (ESBLs) by E. coli represents the primary mechanism responsible for the treatment failure by β-lactam antibiotics [5]. The ESBL-producing E. coli (ESBL-EC) is one of the most important pathogens at the One Health interface [6]. Current ESBLs are mainly encoded by blaCTX-M, blaTEM, blaSHV and blaOXA genes. CTX-Ms are the most predominant types of ESBLs which are encoded by several blaCTX-M type genes [7]) (Tracked Version Line41-48).

Comment 7: In all the manuscript including the abstract the name of bacteria must be in italic

Response 7: Thank you for your comments, we have carefully reviewed the manuscript and italicized the bacterial names.

Comment 8: Please verify the list of references, you need not to write words with capital letters, see the guideline of the journal and a recent article published in the journal. For example change ‘Negeri AA, Mamo H, Gurung JM, Firoj Mahmud AKM, Fällman M, Seyoum ET, et al. Antimicrobial Resistance Profiling and Molecular Epidemiological Analysis of Extended Spectrum β-Lactamases Produced b

---

## [Decision Letter · Decision Letter 1]

3 Mar 2025

PONE-D-24-55572R1Extended-spectrum β-lactamase-producing Escherichia coli isolated from captive primates: characteristics and horizontal gene transfer ability analysisPLOS ONE

Dear Dr. Zhong,

Thank you for submitting your manuscript to PLOS ONE. After careful consideration, we feel that it has merit but does not fully meet PLOS ONE’s publication criteria as it currently stands. Therefore, we invite you to submit a revised version of the manuscript that addresses the points raised during the review process.

We look forward to receiving your revised manuscript.

Kind regards,

Gabriel Trueba, PhD

Academic Editor

PLOS ONE

**Journal Requirements:**

Reviewers' comments:

Reviewer's Responses to Questions

**Comments to the Author**

1. If the authors have adequately addressed your comments raised in a previous round of review and you feel that this manuscript is now acceptable for publication, you may indicate that here to bypass the “Comments to the Author” section, enter your conflict of interest statement in the “Confidential to Editor” section, and submit your "Accept" recommendation.

Reviewer #1: All comments have been addressed

Reviewer #2: All comments have been addressed

2. Is the manuscript technically sound, and do the data support the conclusions?

Reviewer #1: Yes

Reviewer #2: Yes

3. Has the statistical analysis been performed appropriately and rigorously? 

Reviewer #1: Yes

Reviewer #2: Yes

4. Have the authors made all data underlying the findings in their manuscript fully available?

Reviewer #1: Yes

Reviewer #2: Yes

5. Is the manuscript presented in an intelligible fashion and written in standard English?

Reviewer #1: Yes

Reviewer #2: Yes

6. Review Comments to the Author

**Reviewer #1: ** (No Response)

**Reviewer #2:**  Authors have perfectly responded to my previous comments. Just one minor comment to authors, I already sugest its correction but I did not considered a minor correction:

1. Please change '..Those colonies capable of growing on Luria-Bertani (LB) agar supplemented with CTX (4 mg/mL) and sodium aside (100 mg/mL)..' by 'Those colonies capable of growing on Luria-Bertani (LB) agar supplemented with CTX (4 mg/L) and sodium aside (100 mg/L)...'

7. PLOS authors have the option to publish the peer review history of their article (what does this mean? ). If published, this will include your full peer review and any attached files.

**Do you want your identity to be public for this peer review?** For information about this choice, including consent withdrawal, please see our Privacy Policy .

Reviewer #1: No

Reviewer #2: **Yes: ** Mohamed Salah Abbassi

---

## [Author Response · Author response to Decision Letter 2]

4 Mar 2025

Dear Editor:

Thank you very much for your letter and the reviewers’ comments regarding our manuscript submitted to “PLOS ONE” (Manuscript ID: PONE-D-24-55572). We have checked the article and revised it according to the comments, and carefully proof-read the manuscript. The language presentation was improved with assistance from English speakers with appropriate research background. All revisions were marked in revised manuscript with track changes. In addition, in response to latest comments and requirements, we have carefully reviewed all references cited in the manuscript to ensure the completeness and accuracy of the reference list (and checked for any retracted articles as per the journal's guidelines). The additional references added during the previous revision process have been noted in the response letter (Response to Reviewers). We also have made revisions in response to a new comment raised by Reviewer #2.

We hope, with these modifications and improvements based on your suggestion and the reviewer’s comments, the quality of our manuscript would meet the publication standard of “PLOS ONE”. Once again, we acknowledge your comments and constructive suggestions very much, which are valuable in improving the quality of our manuscript.

We submit here the revised manuscript with track changes as well as a list of changes. Thanks again for your reconsideration of our manuscript for publication in your journal. If you have any question about this paper, please don’t hesitate to let me know.

Best wishes for you!

Sincerely yours, Zhijun Zhong

College of Veterinary Medicine

Key Laboratory of Animal Disease and Human Health of Sichuan Province

Sichuan Agricultural University

Chengdu 611130, P. R. China

E-mail: zhongzhijun488@126.com

Response to Reviewer 1 Comments

Response to Reviewer:

Thank you for reading the manuscript carefully and providing many thoughtful comments, which will significantly improve the quality of the manuscript. We have carefully revised the manuscript in accordance with your comments (Tracked Version). Detailed responses are as follows:

Reviewer #1: This manuscript describes the characteristics of Escherichia coli isolated from captive primates. The objective is clear, and the methods used are appropriate. While the results offer some value to readers in this specific field, they do not necessarily present novel findings. I provide the following editorial comments and questions to help improve the readability of the manuscript. Additionally, I strongly recommend that the authors have their revised manuscript undergo thorough English editing before resubmission

Thank you for your suggestions about the language of the manuscript, we carefully checked the English expression and made English revisions under the scrutiny of native English speakers with appropriate research background.

Comment 1: Line 29: remove E. coli

Response 1: Thanks for your comments, we have removed E. coli in our revised manuscript (Escherichia coli is a resident microorganism in the gut of humans and…) (Tracked Version Line 40).

Comment 2: Line 30: Replace all instances of Escherichia coli with E. coli as an abbreviation throughout the entire manuscript.

Response 2: Thanks for your comments, we have checked the abbreviation error and corrected Escherichia coli to E. coli in our revised manuscript.

Comment 3: Line 33: see comment in line 30.

Response 3: Thanks for your comments, we have corrected the error in our revised manuscript (The ESBL-producing E. coli (ESBL-EC) is one of the most important pathogens at the One Health interface…) (Tracked Version Line 45).

Comment 4: Line 46: Remove one occurrence of 'captive' as it is repeated.

Response 4: Thanks for your comments, we have removed the repeated 'captive' in our revised manuscript (In 2024, Bazalar-Gonzales et al [20] isolated seven ESBL-EC strains from captive and semi-captive primates, three blaCTM-X variants...) (Tracked Version Line 62).

Comment 5: Line 76: E. coli should be italicized. This is an incomplete sentence.

Response 5: Thanks for your comments, we have corrected the unitalicized error, checked the grammar and revised the sentence to make the sentence structure complete as follow: ‘Four hundred and forty-four MDR E. coli strains from captive primates, which were isolated from 13 zoos and had previously been stored in our laboratory, were used in this study’ (Tracked Version Line 95-97).

Comment 6: Line 93: please check irp2 or Irp2 here.

Response 6: Thanks for your comments. After checking, we confirmed that it was Irp2 and corrected the error in our revised manuscript.

Comment 7: Line 104: used as the recipient strain

Response 7: Thanks for your comments, we have corrected the error in our revised manuscript (In this study, the sodium azide-resistant E. coli J53 was used as the recipient strain to evaluate the conjugation and transfer ability of…) (Tracked Version Line 129).

Comment 8: Line 105: Those colonies capable….

Response 8: Thanks for your comments, we have added the word to make the sentence correct (Those colonies capable of growing on Luria-Bertani (LB) agar supplemented with CTX (4 mg/mL) and sodium aside (100 mg/mL) were…) (Tracked Version Line 130).

Comment 9: Line 121: This is an incomplete sentence.

Response 9: Thanks for your comments, we carefully checked the sentences for grammatical errors and corrected them as follows: ‘Phylogenetic analysis (A, B1, B2, D) was conducted by detecting yjaA, chuA, and TSPE4.C2, according to Clermont et al’ (Tracked Version Line 150-151).

Comment 10: Line 131: Replace the comma (,) with a semicolon (;).

Response 10: Thank you for pointing out the error, we have corrected it in our revised manuscript (…VAGs were performed using R-software (version 4.2.2) [40]; P-value...) (Tracked Version Line 163).

Comment 11: Line 144-148: You might consider rephrasing it as: Among the five zoos, the most prevalent gene combination pattern in Guiyang Zoo was blaCTX-M-15 (32.00%, 8/25); in Hangzhou Zoo, it was blaCTX-M-55 & blaTEM-198 (34.78%, 8/23); in Chongqing Zoo, it was blaCTX-M-15 (25.00%, 3/12); in Dalian Zoo, it was blaCTX-M-15 (37.50%, 3/8); and in Changsha Zoo, it was blaCTX-M-55 (100%, 1/1).

Response 11: Thank you for the detailed suggestions for revision. We have revised the sentence accordingly in our revised manuscript (Among the five Zoos, the most prevalent gene combination pattern in Guiyang Zoo was blaCTX-M-15 (32.00%, 8/25), in Hangzhou Zoo, it was blaCTX-M-55 & blaTEM-198 (34.78%, 8/23), in Chongqing Zoo, it was blaCTX-M-15 (25.00%, 3/12), in Dalian Zoo, it was blaCTX-M-15 (37.50%, 3/8), and in Changsha Zoo, it was blaCTX-M-55 (100.00%, 1/1).) (Tracked Version Line 183-186).

Comment12: Line 157: replace greatest with largest

Response 12: Thanks for your comments. We have revised this sentence in our revised manuscript.

Comment 13: Line 205-206: Please explain how ESBLs produced by Enterobacteriaceae are ineffective against β-lactam drugs? Do you mean “β-lactam drugs are ineffective against ESBLs produced by Enterobacteriaceae”?

Response 13: Thank you for your comment, ESBLs are produced by some bacteria such as Enterobacteriaceae and are able to hydrolyze β-lactam antibiotics, thus leading to bacterial resistance to these antibiotics. Sorry for the wrong expression here, we have corrected the sentence as follows: "β-lactams are the most prescribed antibiotics worldwide. However, β-lactams antibiotics are ineffective against ESBL-producing Enterobacteriaceae." (Tracked Version Line 253)

Comment 14: Line 211: Only two categories of ESBLs-genes…

Response 14: Thank you for your comment, we have corrected it in our revised manuscript (Only two categories of ESBL genes (blaCTX-M and blaTEM) were…) (Tracked Version Line 259).

Comment 15: Line 216: in a variety of animals

Response 15: Thank you for your comment, we have corrected it in our revised manuscript (blaCTX-M-55 is being increasingly detected in a variety of animals, including waterfowl…) (Tracked Version Line 269).

Comment 16: Line 246-248: You might consider rephrasing it as: The combination of ESBL production and the carriage of virulence-associated genes (VAGs) in E. coli will undoubtedly limit therapeutic options, increase treatment complexity, and enhance pathogenicity.

Response 16: Thank you for your comments. Your opinions make the expression more scientific and rigorous. We have revised it according to your suggestions in our revised manuscript (Tracked Version Line 305-307).

Comment17: Line 248-250: You might consider rephrasing it as: Additionally, strains that carry various VAGs may not be harmful to animals, but there is a potential for them to become pathogenic if host immunity weakens or in response to other adverse environmental factors.

Response 17: Thank you for your comments. We have revised the sentence in our revised manuscript (Tracked Version Line 310-312).

Comment 18: Line 262: The Trbc protein

Response 18: Thanks for pointing out the error, we have corrected it in our revised manuscript (The TrbC protein plays an important role in conjugative transfer…) (Tracked Version Line 330).

Comment 19: Line 267-269: You might consider rephrasing it as: The plasmid incompatibility groups detected in our study included IncFIB, IncFrepB, IncI1, IncFIA, IncHI1, and IncP.

Response 19: Thank you for your comments, we have revised the sentence in our revised manuscript (Tracked Version Line 337).

Comment 20: Line 283-285: You might consider rephrasing it as: Moreover, plasmids, particularly IncF plasmids, have been reported as the most common vehicles for the transmission of VAGs. Our study demonstrated that 9 VAGs can be transferred into the recipient strain J53 through conjugation with plasmids.

Response 20: Thank you for your comments. We have improved the sentence according to the suggestions you provided in our revised manuscript (Tracked Version Line 355-358).

Comment 21: Line 296-300: The ST detected in this study requires a more detailed discussion.

Response 21: Thank you for your comments and we have discussed the detected STs in more detail in our revised manuscript as follow: ‘Moreover, ST533 was previously detected in canine ESBL-EC isolates (harboring the blaCTX-M-15 gene) [73], ST4358 was identified in E. coli isolated from goose feces from China and carried blaCTX-M-55 gene [74], ST683 isolate was previously reported in human carbapenem-resistant E. coli, which was resistant to both cefotaxime and ceftazidime [75]. ST442 has been reported to be associated with Shiga toxin-producing E. coli [76], and ST2161 has been reported in fosfomycin-resistant E. coli from humans [77]. As far as we know, ST469, ST2854, and ST4040 have not been reported in ESBL-EC. Our results indicated that ESBL-EC isolates from captive primates exhibited diverse clonal lineages, which may be attributed to cross-infections among different animal species, as well as between humans and captive primates.’ (Tracked Version Line 378-387).

Added references:

73. Li S, Liu J, Zhou Y, Miao Z. Characterization of ESBL-producing Escherichia coli recovered from companion dogs in Tai'an, China. Journal of infection in developing countries. 2017;11(3):282-6.

74. Cen DJ, Sun RY, Mai JL, Jiang YW, Wang D, Guo WY, et al. Occurrence and transmission of bla(NDM)-carrying Enterobacteriaceae from geese and the surrounding environment on a commercial goose farm. Appl Environ Microbiol. 2021;87(10).

75. Tsilipounidaki K, Florou Z, Skoulakis A, Fthenakis GC, Miriagou V, Petinaki E. Diversity of bacterial clones and plasmids of NDM-1 producing Escherichia coli clinical isolates in central Greece. Microorganisms. 2023;11(2).

76. Fierz L, Cernela N, Hauser E, Nüesch-Inderbinen M, Stephan R. Characteristics of shigatoxin-producing Escherichia coli strains isolated during 2010-2014 from human infections in Switzerland. Front Microbiol. 2017;8:1471.

77. Grilo T, Freire S, Miguel B, Martins LN, Menezes MF, Nordmann P, et al. Occurrence of plasmid-mediated fosfomycin resistance (fos genes) among Escherichia coli isolates, Portugal. Journal of global antimicrobial resistance. 2023;35:342-6. 

Response to Reviewer 2 Comments

Response to Reviewer:

Thank you for reading the manuscript carefully and providing many thoughtful comments, which will significantly improve the quality of the manuscript. We have carefully revised the manuscript in accordance with your comments (Tracked Version). Detailed responses are as follows:

Reviewer #2:

Comment 1: In the abstract you have to mention the studied samples, their number, their origins, region of the study. The aim not mentioned, the experiment were not mentioned. Perhaps you are limited by the number of words that must in the abstract but few data are needed to understand the work by learning the abstract.

Response 1: Thanks for your comments, we have improved the abstract section. We have added research objectives, sample information such as sources, quantities, and experimental methods to the abstract in our revised manuscript (The rapid spread of extended-spectrum β-lactamases (ESBLs)-producing Escherichia coli (ESBL-EC) around the world has become a significant challenge for humans and animals. In this study, we aimed to examine the characteristics and horizontal gene transfer (HGT) capacity of ESBL-EC derived from captive primates. We screened for ESBL-EC among a total of 444 multidrug-resistant (MDR) E. coli strains isolated from 13 zoos in China using double-disk test. ESBL genes, mobile genetic elements (MGEs), and virulence-associated genes (VAGs) in ESBL-EC were detected through polymerase chain reaction (PCR). Furthermore, conjugation experiments were conducted to examine the HGT capacity of ESBL-EC, and the population structure (phylogenetic groups and MLST) was determined.) (Tracked Version Line 6-12).

Comment 2: Abstract, line 19, write ‘...The population structure analysis showed that the phylogroup B1 and ST2161 were the most prevalent...’

Response 2: Thanks for your comment, we have corrected the sentence in the abstract section in our revised manuscript (Tracked Version Line 26-27).

Comment 3: Line 20 you wrote ‘ESBL-EC derived from captive primates pose a potential threat to animals, the environment, and humans through horizontal gene transfer.’ In reality the threat is by direct contact with colonized primates or by environmental pollution, so horizontal transfer is not a strict threat, perhaps genetic transfer is more implicated in the transfer between several genera colonizing primate. Take this in mind and try to modify this sentence in the abstract and this idea in the remaining part of your manuscript.

Response 3: Thank you for your comments. We have revised the manuscript to make the expression correct and scientific in our revised manuscript. Our revisions are as follows: ESBL-EC poses a potential threat to captive primates and may spread to other animals, humans, and the environment. It is imperative to implement measures to prevent the transmission of ESBL-EC among captive primates (Tracked Version Line28-30); ESBL-EC isolates may pose a serious threat not only to captive primates but also to other animals, humans, and the environment through cloning and direct contact (Tracked Version Line363-365).

Comment 4: In general we say ‘Mobile genetic element(s) (MGE)’ not ‘Mobile gene element (MGE)’ so modify this in the keywords and in the manuscript.

Response 4: Thanks for your comment, we have corrected the problem and made corrections to the errors elsewhere in our revised manuscript.

Comment 5: Write ‘Keywords: Captive primate; Escherichia coli; Extended‐spectrum β‐lactamase (ESBL), Virulence-associated gene (VAG); Mobile genetic elements (MGEs); Horizontal gene transfer (HGT).

Response 5: Thanks for your comments, we have adjusted the order of keywords in our revised manuscript (Keywords: Captive primate; Escherichia coli; Extended‐spectrum β‐lactamase (ESBL); Virulence-associated gene (VAG); Mobile genetic element (MGE); Horizontal gene transfer (HGT)) (Tracked Version Line34-37).

Comment 6: Line 30, write ‘...Antimicrobial resistant (AMR) E. coli poses a threat to animal............ The production of extended-spectrum β-lactama

---

## [Decision Letter · Decision Letter 2]

7 Mar 2025

Extended-spectrum β-lactamase-producing Escherichia coli isolated from captive primates: characteristics and horizontal gene transfer ability analysis

PONE-D-24-55572R2

Dear Dr. Zhong,

We’re pleased to inform you that your manuscript has been judged scientifically suitable for publication and will be formally accepted for publication once it meets all outstanding technical requirements.

Kind regards,

Gabriel Trueba, PhD

Academic Editor

PLOS ONE

Additional Editor Comments (optional):

Reviewers' comments:

Reviewer's Responses to Questions

**Comments to the Author**

1. If the authors have adequately addressed your comments raised in a previous round of review and you feel that this manuscript is now acceptable for publication, you may indicate that here to bypass the “Comments to the Author” section, enter your conflict of interest statement in the “Confidential to Editor” section, and submit your "Accept" recommendation.

Reviewer #2: All comments have been addressed

2. Is the manuscript technically sound, and do the data support the conclusions?

Reviewer #2: Yes

3. Has the statistical analysis been performed appropriately and rigorously? 

Reviewer #2: Yes

4. Have the authors made all data underlying the findings in their manuscript fully available?

Reviewer #2: Yes

5. Is the manuscript presented in an intelligible fashion and written in standard English?

Reviewer #2: Yes

6. Review Comments to the Author

Reviewer #2: Dear authors

Authors have perfectly responded to my previous comments. I have not other comments. Also it seems that comments from other reviewers have enhenced the quality of the article.

7. PLOS authors have the option to publish the peer review history of their article (what does this mean? ). If published, this will include your full peer review and any attached files.

**Do you want your identity to be public for this peer review?** For information about this choice, including consent withdrawal, please see our Privacy Policy .

Reviewer #2: **Yes: ** Mohamed Salah Abbassi

---

## [Editor Report · Acceptance letter]

PONE-D-24-55572R2

PLOS ONE

Dear Dr. Zhong,

I'm pleased to inform you that your manuscript has been deemed suitable for publication in PLOS ONE. Congratulations! Your manuscript is now being handed over to our production team.

Kind regards,

on behalf of

Dr. Gabriel Trueba

Academic Editor

PLOS ONE